# The Effect of Internal and external imagery on learning badminton long serve skill: The role of visual and audiovisual imagery

**Fateme Parimi** [1] *, **Behrouz Abdoli**[2], **Hesam Ramezanzade**[3], **Mahin Aghdaei**[2]

1 Ms in Learning and Motor Control, Department of Cognitive and Behavioral Sciences and Technology in Sport, Shahid Beheshti University, Tehran, Iran, 2 Department of Cognitive and Behavioral Sciences and Technology in Sport, Shahid Beheshti University, Tehran, Iran, 3 Department of Sport Science, School of Humanities, Damghan University, Damghan, Iran

* fateme.parimy@gmail.com

**Data Availability Statement:** All relevant data are within the manuscript and its Supporting Information files.

## Abstract

This study aimed to examine the impact of internal and external audiovisual imagery on the learning of the badminton long serve skill. A lot of 42 right-handed novice women were selected using availability sampling. Participants were categorized into four groups based on their scores from the visual imagery ability questionnaire and Bucknell auditory questionnaire: Visual-Internal imagery, Visual-External imagery, AudioVisual-Internal imagery and AudioVisual-External imagery groups. To generate an auditory pattern, the shoulder joint's angular velocity of a skilled individual was recorded and translated into sound based on frequency characteristic changes. Subjects underwent four sessions of 40 trials each and subsequently participated in retention and transfer tests. Performance accuracy of the badminton long serve was assessed using the Scott and Fox standard test and repeated measures ANOVA was employed to compare performance across groups during test stages. While no significant differences were noted between groups during the acquisition stages, indicated that subjects in the AudioVisual imagery conditions outperformed those in Visual imagery during the retention test. Additionally, the AudioVisual-Internal Imagery group demonstrated superior performance compared to other groups. Internal imagery groups also exhibited better performance in the later stages of acquisition, retention and transfer tests compared to external imagery groups. These findings suggest that the incorporation of audiovisual imagery utilizing movement sonification, alongside physical practice, improves skill development more effectively than visual imagery alone.

## 1. Introduction

Learning is characterized as "relatively stable changes in a person's behavior". It is a complex process that occurs in the nervous system and is not directly observable. Learning can be examined from various angles. Motor learning is a key aspect, referring to a series of processes associated with practice or experience that result in a lasting change in the ability to perform

**Funding:** The author(s) received no specific funding for this work.

**Competing interests:** The authors have declared that no competing interests exist.

motor skills [1]. The learning of motor skills is greatly influenced by the type of education and practice method. Therefore, there are various methods for educating and practicing motor skills that can improve the learning process and ultimately improve the quality of practice. One effective method commonly used in educational settings is the use of mental imagery. Studies in motor behavior sciences and sports psychology have demonstrated that cognitive interventions such as imagery, along with physical practice, facilitate motor learning [2–4]. Neurological studies suggest that during imagery and mental performance of a skill, the same neural and cognitive operations are activated as during physical practice [5, 6]. Little wonder, then, that recent studies have highlighted the positive role of mental imagery in learning and performing motor skills, based on theories and hypotheses proposed in the field of imagery including the Psycho Neuromuscular Theory, a which posits that mental imagery creates similar neural mechanisms to those during physical practice [4, 7]. According to symbolic learning theory, imagery assists subjects in reviewing the sequence of movements as symbolic components of a task. This increased focus on key task components through imagery contributes to the construction of movement schemas in the primary motor cortex. Additionally, Paivio (1971) stated in the Dual Code Theory that imagery is effective in learning by creating two independent memory codes. It is possible to learn sequences of movements both verbally and visually. Research indicates that these two types of coding are independent, meaning we can forget one code without affecting the memory of the other code.

The majority of research in learning and motor control has concentrated on single sensory dimensions, particularly emphasizing the visual sense. Likewise, studies in the realm of imagery have predominantly centered on visual imagery alone. However, Vealey and Greenleaf [8] define mental imagery as a conscious experience that engages all senses to recreate an event in the mind. According to this broad definition, imagery encompasses various sensory modalities, including visual, kinesthetic, auditory, tactile, olfactory, and taste imagery. Among these sensory aspects, visual imagery has garnered significant interest, with most studies in the field focusing on visual imagery to explore its impact on performance and learning [9–12]. Visual imagery comprises two types: internal and external. Internal imagery involves envisioning the performance from a first-person perspective, akin to what one sees during actual execution. In contrast, external imagery entails observing one's performance from a third-person perspective. Despite the prominence of visual imagery, other sensory aspects such as auditory imagery have received comparatively less attention. Although some studies suggest the utility of auditory imagery, it remains understudied relative to visual imagery. Auditory imagery involves mentally organizing and analyzing sounds when no auditory stimulus is present, constructing a mental representation of an auditory event [13].

Intons-Peterson [14] defined auditory imagery as "the permanence of an auditory experience made up of its components in long-term memory, in the absence of direct sensory stimulation of that experience." Smith and Holmes [15] demonstrated that utilizing auditory imagery, such as one's recorded movement sound, improved putting performance in golfers. Additionally, Debarnot and Guillot [16] found that presenting rhythmic motor-related music during imagery practice enhanced temporal coordination between physical and imagery practice, thereby boosting efficacy. Pourmorad Kohan et al. [13] examined the effect of visual, auditory, and kinesthetic imagery on learning basketball shooting skills but found no significant difference among experimental groups. This aligns with Pizzamiglio et al. [17] findings that non-action-related sounds fail to activate the motor and premotor brain areas. In auditory imagery studies, researchers have predominantly utilized sounds from the environment, coaches, or spectators [18], overlooking motor pattern-related sounds. Some studies, the recorded natural sound of the individual's own performance has been used [15, 19, 20]. Because these natural movement sounds may not represent the motor pattern, in recent

researches, the change of kinetic or kinematic parameters related to the motor pattern (which cannot be heard under normal conditions) has been presented to individuals in the form of sound [21–24]. These sounds related to motor are known as sonification. Sonification is a concept used to convert kinetic or kinematic characteristics such as displacement or velocity of one's motor into auditory patterns, which according to studies can be effective in developing performance and learning [25–30].

Several researchers have demonstrated that combining auditory models (sonification) with visual cues influences movement perception [22, 24, 29, 31, 32]. For instance, Effenberg et al. showed the impact of sonification on learning closed motor skills (indoor rowing technique acquisition) [28]. Ramzanzade et al., in studies on multisensory perception, demonstrated how audiovisual integration affects performance accuracy and motor learning, illustrating the perceptual-motor enrichment effect [30, 33, 34].

Despite evidence of the positive effects of sound-based interventions (sonification) on perception and motor learning, their influence on mental imagery remains unclear. Existing studies have largely employed environmental sounds or music in auditory imagery [13, 15, 18–20] and have predominantly explored imagery in a single-sensory manner. However, some studies have utilized audiovisual imagery based on sonification. Castro et al. [35] found that sonification-based imagery did not affect corticospinal excitability during simple tasks. Conversely, Ramzanzade et al. [36] demonstrated that sonification-based audiovisual imagery increased muscle activity amplitude.

Given the range of muscle activity affected by audiovisual imagery in Ramzanzade's study, it prompts investigation into whether this effect might also influence motor skill acquisition and learning. The auditory pattern used in this research, combined with visual cues, facilitates the mental rehearsal of movement sequences as symbolic task components (symbolic learning theory). Furthermore, it is anticipated that this auditory pattern aids in subsequent movement sequence recall by establishing an independent memory code (dual coding theory).

Our specific hypotheses were as follows:

***Hypothesis 1***: Building upon prior research [36] and grounded in symbolic learning and dual coding theories, we hypothesized that audiovisual imagery would be both more likely to occur and more effective compared to visual imagery in facilitating the learning of the badminton long-serve skill.

***Hypothesis 2***: Given that the auditory pattern utilized in this study (sonified angular velocity of the shoulder joint) provides direct information about the movement pattern itself, we predicted that it would enhance the sense of kinesthetic awareness and be particularly effective under conditions of internal imagery.

## 2. Research method

This study used a semi-experimental design with a pre-test and post-test structure, employing Available sampling. The participant pool consisted of 42 female students aged between 18–23 years who volunteered for the research.

The G * power software was used to calculate the sample size for the ANOVA repeated measures test ($\alpha = .05$; $\beta = .95$, group number = 4, number of measurement = 5, effect size = .223), one way ANOVA and two way ANOVA ($\alpha = .05$; $\beta = .85$, group number = 4, effect size = .58) [13, 15]. All participants were right-handed and free from neurological, hearing, visual, motor, attention and depression disorders, verified through self-report assessment. All participants provided written consent to participate in the research. The study protocol was approved by the Ethics Committee of Shahid Beheshti University (ethics code: IR.SBU. REC.1401.116).

## 2.1. Measuring tools

To collect behavioral data, the mental rotation test (MRT), visual imagery ability questionnaire (internal, external, kinesthetic) and Bucknell's auditory imagery scale (vividness and control) were used. Also, Scott and Fox's standard test with validity 0.54 and reliability 0.70 was used to record the performance results.

**2.1.1. Mental Rotation Test (MRT).**   Mental rotation refers to the ability to rotate the mental representations of two-dimensional and three-dimensional objects, which is attributed to the visual representation of such rotation in the human mind [37]. Mental rotation is a cognitive function that helps a person comprehend what has changed in the state of the object. The mental rotation test consists of numbers presented by Shephard and Metzler (1971), which is based on a version of AutoCAD drawing and the mental rotation test of Vanderberg and Kius [37]. There are two types of mental rotation test called V, 20 questions and K, 24 questions. In this research, a 24-question mental rotation test was used. Each problem consists of a target shape on the right and four stimulus shape on the left. Two of the four stimulus shapes are the rotated version of the target shape and the other two shapes are not identical to the target shapes. This test was conducted on a group of students other than the sample of the main research in a preliminary research and the reliability of the test was calculated using the test-retest method as 0.87. A cut-off point of 70% was used to select subjects to participate in the research.

**2.1.2. Visual imagery ability questionnaire (external, internal, kinesthetic) (MIQ-3).** This questionnaire, which evaluates visual imagery ability was first presented by Isak, Marks and Russell (1986) to fill the gap in the literature related to motor imagery. The third version of this questionnaire was presented in 2012 by Wilson et al. They designed a modified version of this questionnaire called Visual Imagery 3 (MIQ-3) in order to accurately measure visual imagery components (internal and external) and overcome other shortcomings related to the questionnaire (MIQ-R). This questionnaire evaluates internal, external, and visual imagery. This 24-question questionnaire was measured with a 5-point Likert scale. There were 8 questions to measure the imagery ability of each type of imagery (external, internal, kinesthetic). The results of Wilson et al. (2012) confirmed that this questionnaire has very good construct validity. They also reported good validity for the subscales of this questionnaire including internal visual imagery (0.628), external visual imagery (0.679) and kinesthetic imagery (0.706). Badiei (2013) evaluated the validity and reliability of the motor imagery vividness questionnaire in Iran and found that the questionnaire questions accounted for about 47% of the total variance related to visual, internal and motor imagery vividness and the overall scale, respectively 0.86 and 89. 0, 0.91 and 0.95, which shows the high reliability of this subscale and the overall scale of motor imagery vividness. Also, exploratory factor analysis was used for the construct validity of the questionnaire.

**2.1.3. Bucknell auditory imagery scale (vividness and control).**   The Bucknell Auditory Imagery Scale (BAIS) is a short self-report scale that includes three main dimensions of auditory experience: musical, verbal, and environmental sounds. For each item in BAIS, a situation is described and then the sound associated with that situation is presented. The subjects are required to make an auditory image of the presented sound. For the vividness subscale (BAIS-V), the subjects are asked to rate their image vividness of sound using a scale of 1 to 7, where one means no image, four means relatively clear, and seven means that the image is as clear as real. No other scale points are labeled. For the control subscale (BAIS-C), each situation and sound is described once again, but the purpose of the rating is to facilitate the image change from the original sound to the new sound. Subjects use a different 7-point scale to rate control, where one means no image, four means being able to change the image, but with

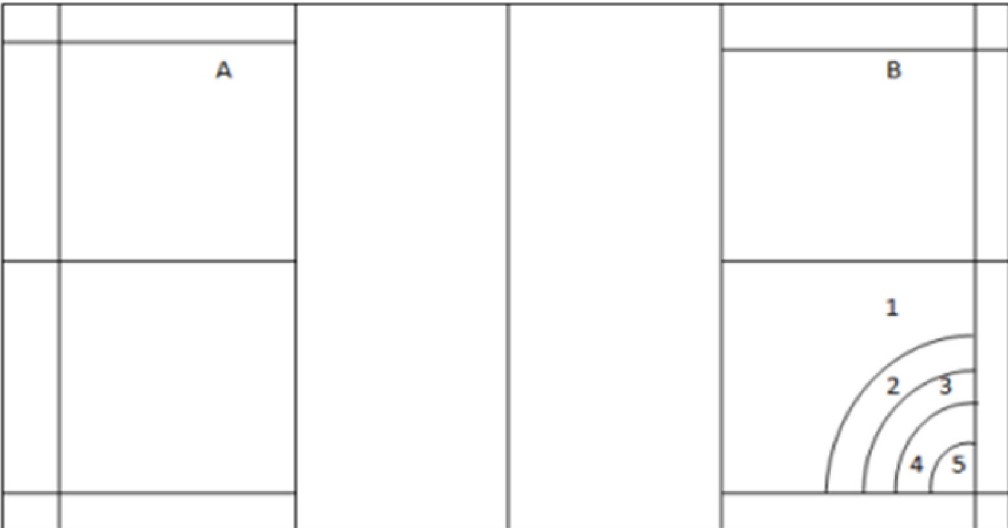

**Fig 1. Scott and fox badminton long serve standard test.**

effort, and 7 means changing the image very easily. The face and content validity of the Bucknell auditory imagery questionnaire (vividness and control) was confirmed by experts with a content validity coefficient of 0.72.

**2.1.4. Scott and Fox badminton long serve standard test.** To implement this test, the end of the badminton court was scored with five concentric semi-circles, and based on the landing of the serve hit on each of the designated areas, the score corresponding to that area was recorded for the individuals. The distance between each semi-circle was five centimeters and each semi-circle was given 1, 2, 3, 4, 5 points respectively (Fig 1).

# 3. Audiovisual pattern

Initially, a skilled individual with over 10 years of experience performed the badminton long serve, which was recorded using motion analysis equipment equipped with IMU sensors (Inertial Measurement Unit) placed on the arm, forearm, and mid-body. Two cameras were positioned to capture the performance from medial and lateral views (angled at 30 degrees to the sagittal plane). The angular velocity of the shoulder joint during the serve was captured and converted into sound using Sandbox sonification software (Version 6), adjusting the frequency to create an auditory pattern. Subsequently, the auditory pattern was synchronized with the visual performance to create an audiovisual pattern [38].

# 4. Research methodology

This research was conducted in the form of pre-test, practice sessions, post-test, retention test and transfer test. At the beginning and before the pre-test, the subjects completed personal information form and the consent form indicating voluntary participation in the study. After that, to assess the imagery ability of the subjects, they first participated in the mental rotation test and those right-handed and without neurological, hearing, visual, motor, attention and depression disorders who scored above 70% were selected to participate in the research. And after receiving full explanations from the examiner regarding the types of imagery and their differentiation, in order to match the groups in terms of visual and auditory imagery ability,

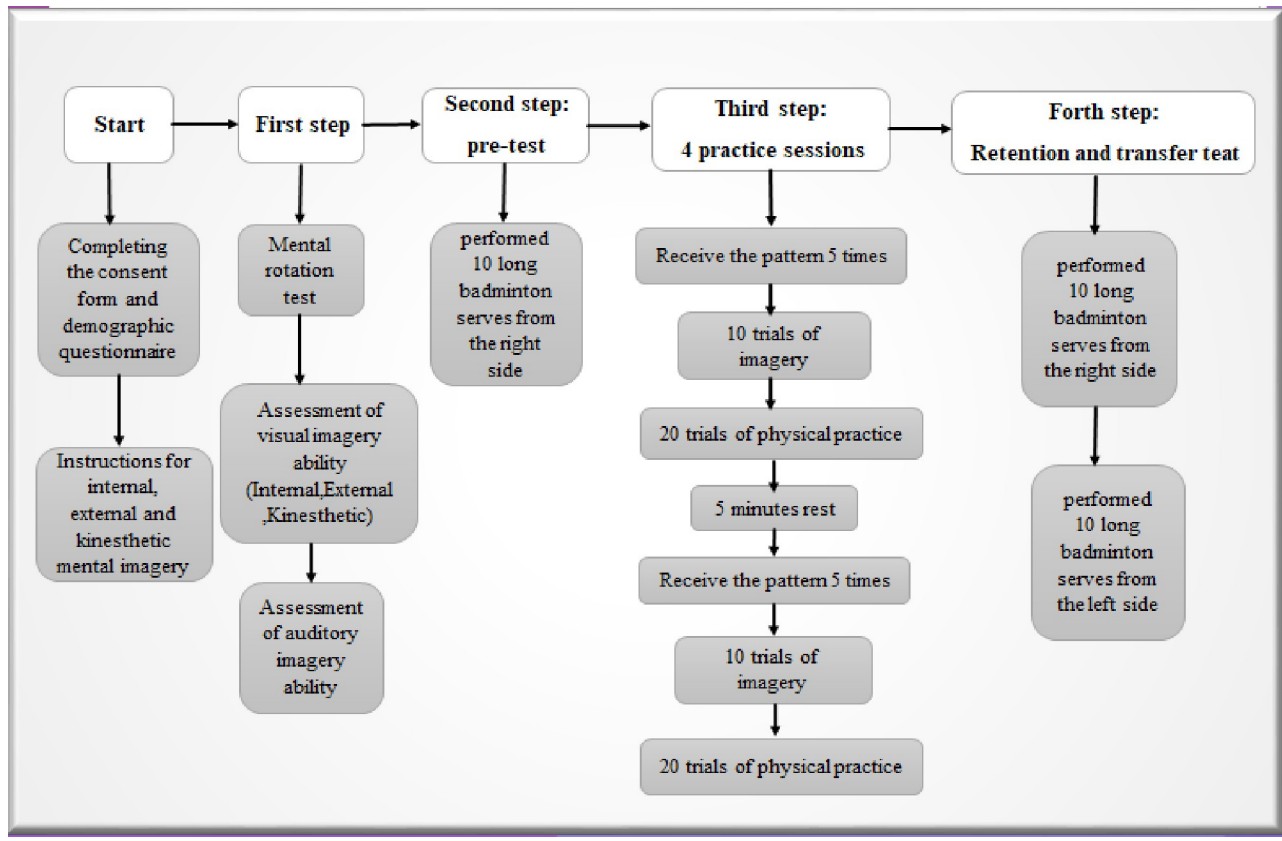

**Fig 2. Overview of research methodology.**

the subjects completed the following questionnaires: MIQ imagery ability and Bucknell auditory imagery vividness and control questionnaire.

In the pre-test phase, participants performed 10 badminton long serves from the right side of the court, with performances recorded for analysis. Four groups were established: Visual-Internal imagery, Visual-External imagery, AudioVisual-Internal imagery, and AudioVisual-External imagery. Each group underwent four practice sessions comprising 40 trials.

During practice sessions, each group received their respective pattern (visual/audiovisual) followed by 10 imagery trials specific to their group perspective (internal/external) and 20 physical performances. Post-session, participants rested for 5 minutes before repeating the pattern exposure, imagery trials, and physical performance sequence. Following each practice session, participants performed 10 serves for scoring.

Retention and transfer tests occurred 48 hours after the final practice session. The retention test mirrored the pre-test conditions, evaluating serve performance. A transfer test followed, assessing performance with serves from the left side of the court and the scores were recorded based on the Scott and Fox standard tests (Fig 2).

## 4.1. Statistical methods

Performance scores were described using mean and standard deviation. The normality of data distribution was assessed using the Shapiro-Wilk test, and the homogeneity of variances was tested using Levene's test. Repeated Measures ANOVA was employed to compare groups across acquisition stages (4 groups * 5 stages). One-way ANOVA and two-way ANOVA were

used to compare groups in retention and transfer tests. Specifically, one-way ANOVA analyzed group performance in pre-test, retention, and transfer tests, while two-way ANOVA evaluated the interactive effect of imagery type (visual, audiovisual) and perspective (internal, external) in retention and transfer tests.

## 5. Research findings

Initially, the study compared the visual and auditory imagery abilities of subjects across four research groups during a pre-test phase. The results revealed no significant differences in the level of visual and auditory imagery ability among these groups.

For analyzing the performance during the acquisition sessions, Repeated Measures ANOVA was used. Due to the violation of the assumption of homogeneity of variance-covariance (Mauchly's W = 0.516), the Greenhouse-Geisser method was utilized. The findings indicated that the interaction effect $(F_{9.466, 119.905}) = 0.787$, p = 0.635, Eta = 0.059) and the main effect of group $(F_{3, 38}) = 0.503$, p = 0.683, Eta = 0.038) were not significant, implying no significant differences in accuracy among the groups across the acquisition phases. However, a significant main effect of the test was observed $(F_{4, 152}) = 13.16$, p < 0.001, Eta = 0.257), signifying differences across different stages of the test. Bonferroni's post hoc test indicated significant improvements in performance from the pre-test to the third session (MD = 0.31, p = 0.001, Cohen's d = 0.806) and the fourth session (MD = 0.379, p = 0.001, Cohen's d = 1.338). Additionally, performance in the first session significantly differed from that in the third (MD = 0.236, p = 0.018, Cohen's d = 0.520) and fourth sessions (MD = 0.305, p = 0.001, Cohen's d = 0.829).

Fig 3 illustrates the performance of the subjects in the four groups during retention and transfer tests. The AudioVisual-Internal imagery group performed notably better in both retention and transfer tests.

The results of the retention test demonstrated a significant group effect $(F_{3, 38}) = 4.315$, p = 0.01, Eta = 0.254), indicating differences among the groups. Bonferroni's post hoc analysis revealed significant differences between the performance of the AudioVisual-Internal group and the Visual-Internal (MD = 0.236, p = 0.049, Cohen's d = 1.643), Visual-External (MD = 0.418, p = 0.001, Cohen's d = 1.755), and AudioVisual-External groups (MD = 0.278, p = 0.025, Cohen's d = 0.914), all in favor of the AudioVisual-Internal group. No significant differences were observed among the other groups.

Based on the results of the transfer test, a significant group effect was observed $(F_{3, 38}) = 4.499$, p = 0.009, Eta = 0.262), indicating differences among the research groups in the transfer test. Bonferroni's post hoc test identified significant differences between the performance of the AudioVisual-Internal group and the Visual-External (MD = 0.299, p = 0.029, Cohen's d = 1.502) and AudioVisual-External groups (MD = 0.289, p = 0.038, Cohen's d = 1.213), favoring the AudioVisual-Internal group. However, no significant difference was observed between the AudioVisual-Internal and Visual-Internal groups.

Fig 4 illustrates the performance comparison between Visual imagery and AudioVisual imagery groups in the retention and transfer tests. It is evident from Fig 3 that the AudioVisual imagery groups outperformed the Visual imagery groups in the retention test.

Fig 5 depicts the performance comparison between Internal and External imagery groups in the retention and transfer tests. As shown in Fig 4, the internal imagery groups exhibited superior performance compared to the external imagery groups in both retention and transfer tests.

The results of two-way ANOVA for the retention test revealed significant main effects of imagery perspective $(F_{1, 38}) = 7.476$, p = 0.009, Cohen's d = 0.798) and imagery sense $(F_{1, 38})$

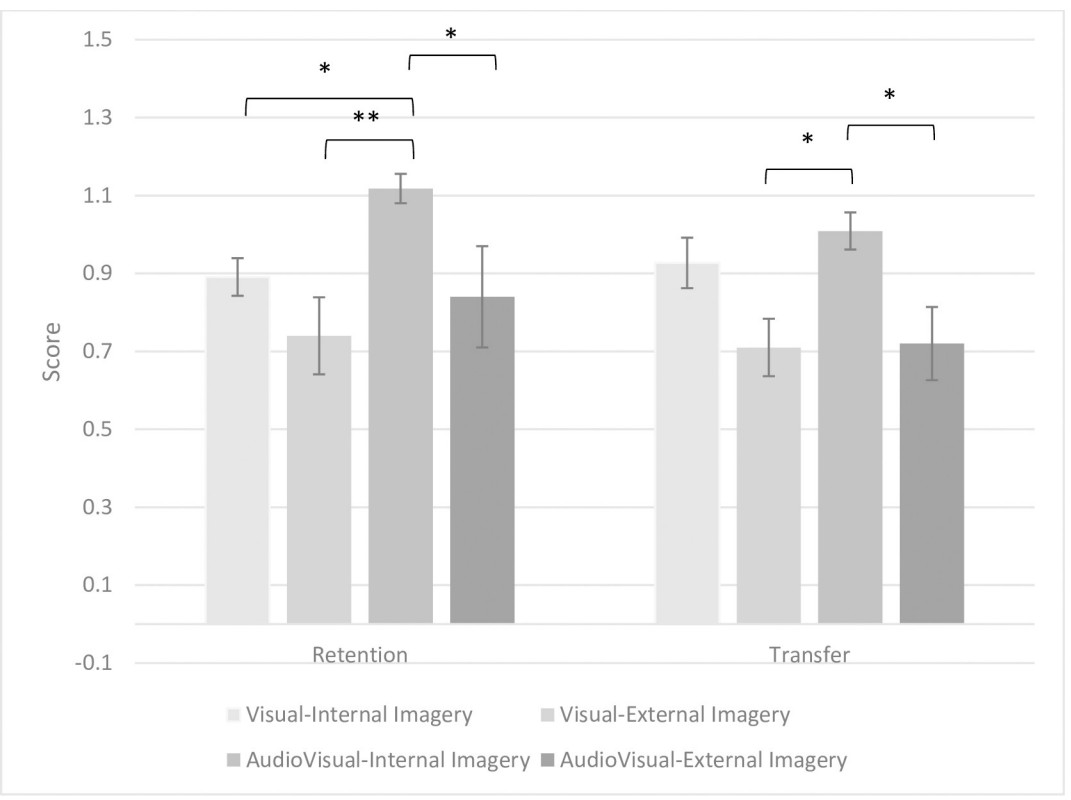

**Fig 3. Performance of the subjects in four research groups in retention and transfer tests.** * p < 0.05, ** p < 0.01.

= 5.005, p = 0.031, Cohen's d = 0.655). These findings underscore that regardless of the imagery sense (Visual or AudioVisual), there is a significant advantage for internal imagery over external imagery. Additionally, the results showed a significant difference between visual and audiovisual imagery, favoring audiovisual imagery, irrespective of the imagery perspective (internal or external).

In the transfer test, the two-way ANOVA revealed a significant main effect of imagery perspective (F1, 38) = 12.785, p = 0.001, Cohen's d = 1.113), suggesting that regardless of the imagery sense (visual or audiovisual), internal imagery yields better performance than external imagery. However, the main effect of imagery sense was not significant (F1, 38) = 0.420, p = 0.521, Cohen's d = 0.181), indicating no significant difference between visual and Audio-Visual imagery in the transfer test, irrespective of the imagery perspective (internal or external).

## 6. Discussion and conclusion

The objective of this study was to evaluate the impact of internal and external audiovisual imagery on the performance of the badminton long serve across the stages of acquisition, retention, and transfer. The findings revealed that while all groups showed improvement during the acquisition sessions, there was no significant difference between groups in terms of execution accuracy during this phase. However, the AudioVisual-Internal group exhibited a notable advantage over other groups in the retention test.

The lack of significant differences in execution accuracy during acquisition could be attributed to the increased cognitive workload experienced by the AudioVisual groups. In the

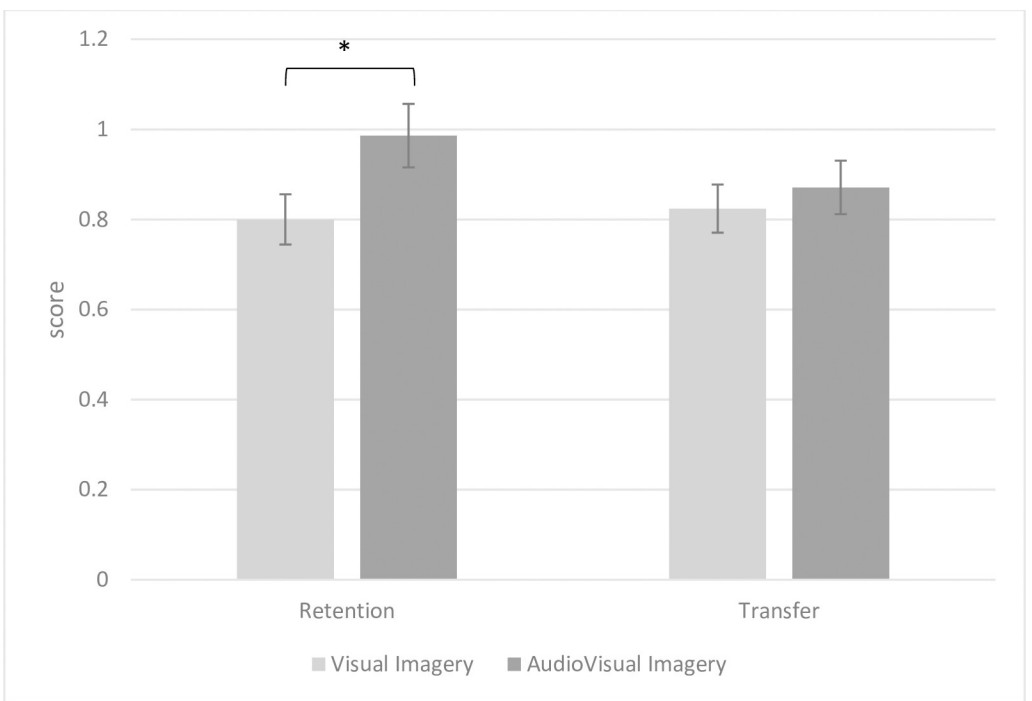

**Fig 4. Performance comparison of visual imagery and audiovisual imagery groups in retention and transfer tests.** $^*$ $p < 0.05$.

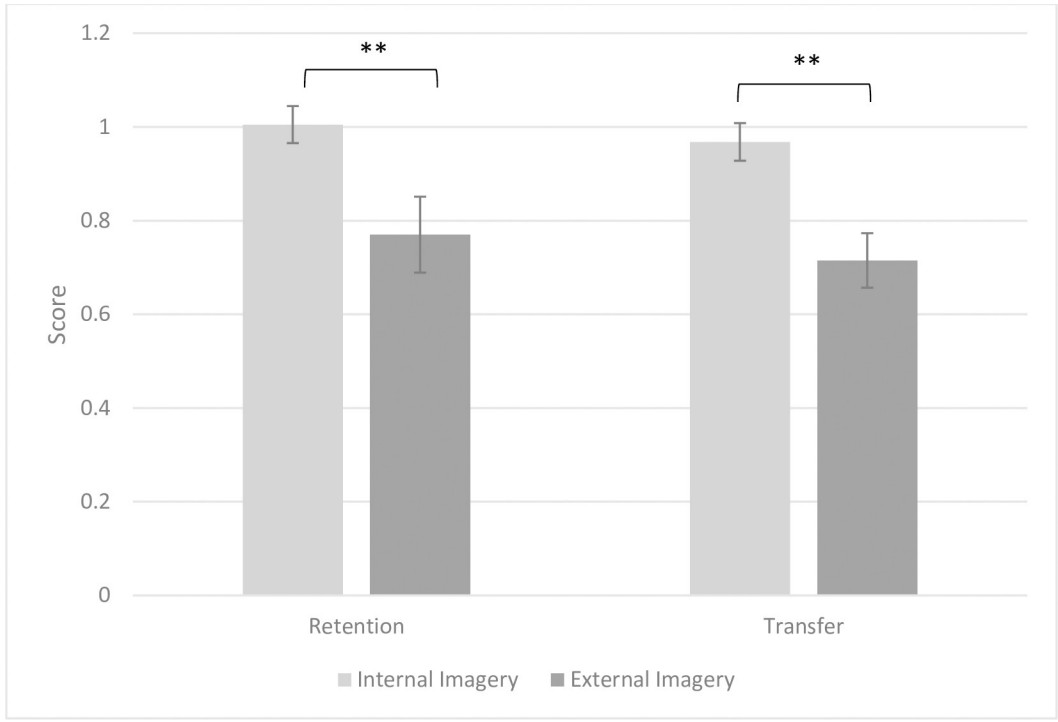

**Fig 5. Performance comparison of internal and external imagery groups in retention and transfer tests.** $^{**}$ $p < 0.01$.

acquisition stage, participants in these groups needed to process both visual and auditory patterns simultaneously, integrating them during the cognitive phase of learning. This increased demand likely overwhelmed working memory capacity. Previous studies by Schmidt et al. [39] and Hasting & West [40] have demonstrated that working memory, essential for complex cognitive tasks, can negatively impact performance under conditions of overload. Furthermore, the introduction of unfamiliar sonified sounds in the audiovisual stimulus may have drawn excessive attention, diverting cognitive resources away from performance analysis. Kanmen [41] suggests that novel stimuli can pose cognitive challenges, particularly affecting individuals' ability to process information optimally. For beginner-level participants, the heightened challenge posed by the amount and complexity of information in the audiovisual pattern likely hindered optimal performance during practice sessions [42].

Generally speaking, in practice sessions, performance can often misrepresent learning due to the influence of various performance variables and the presence of performance plateaus.

The results observed during the acquisition stages can be elucidated using the motor behavior-memory framework [43]. According to this framework, a distinction between performance and learning emerges, particularly evident when challenging practice conditions may hinder immediate performance but improve long-term retention of motor skills. This framework underscores three key memory processes (encoding, consolidation, and retrieval) that operate over the course of behavior, reflected in the stages of practice, post-practice interval, and long-term retention. During the acquisition phase, encoding processes predominate as the learner practices the motor skill. This phase involves cognitive processes such as stimulus identification, response selection, and execution, which may be influenced by cognitive interventions similar to those employed in the present study. The impact of these cognitive processes during encoding can lead to performance differences observed in subsequent retention tests, where motor memory is stabilized. Therefore, retention and transfer tests can be more effective means of assessing learning [1]. It is recommended that the outcomes of this study be complemented with performance process assessments (e.g., kinematic analysis) during the acquisition phase. This approach can help differentiate interventions made during trial and error for beginner subjects, highlighting changes in the performance process rather than solely focusing on outcomes. It has also been demonstrated that the effects of cognitive interventions on performance may manifest with a delay compared to motor interventions. Considering the improvement observed in the audiovisual groups during practice sessions (see Fig 2), there is a possibility and guesswork that continued practice sessions may reveal distinct performance differences between groups as a consequence of interventions implemented during the acquisition phases.

In the retention test, a significant and sizeable effect favoring AudioVisual imagery over Visual imagery groups was observed. This finding suggests that reducing cognitive load during the retention test, achieved by removing the pattern observation and subsequent imagery requirements, allowed subjects in the AudioVisual groups to perform better than those in the Visual imagery groups. This effect can be attributed to the multi-sensory practice efforts during the acquisition sessions involving audiovisual imagery. Participants in the AudioVisual group, while initially focusing internally during acquisition, likely implicitly acquired information through consistent exposure to models and imagery, leading to improved performance in the retention test when pattern and imagery were removed [43, 44]. Studies by Schmidt et al. [45], indicate that motor sonification improves activity in the action observation system, including subcortical motor loop structures. Multi-sensory integration mechanisms compensate for visual information deficits by utilizing the auditory sense when visual information is insufficient [46]. Schmidt et al. [45] also showed that regions such as the superior temporal sulcus (STS) and mirror neurons involved in visual perception are activated during auditory

perception with sounds obtained from motor itself as part of audiovisual integration and facilitates learning processes. Research shows that auditory mirror neurons are activated only when sounds related to the kinematics of the motor pattern are used [17]. According to modeling theory, imagery is a form of modeling where individuals observe and implement modeled behavior themselves. Gibson's theory of direct perception suggests that audiovisual imagery allows direct reception of specific movement-related information, facilitating learning. Attention to sonified sounds likely helps movement detail retention and prevents forgetting, influencing learning outcomes. Additionally, hearing is more accurate and precise than vision in timing perception, making it ideal for perceiving joint angle velocity. Auditory imagery preserves auditory stimuli characteristics, aiding motor timing [14, 47, 48]. In addition, sound is effective in perceiving the order of movement.

Multisensory stimuli are processed more efficiently and accurately than unisensory stimuli, improving the reception and quality of mental images with audiovisual convergence. Recent studies demonstrate higher muscle activity intensity during audiovisual imagery compared to visual imagery, indicating a stronger effect of this type of imagery [36]. Previous research highlighting the benefits of auditory imagery often used music or sport-related sounds [15, 48] Suggest a relationship between motor timing prediction and musical imagery ability, supporting the notion that simultaneous visual and auditory pattern exposure during audiovisual imagery improves motor pattern timing in mental images, thereby enhancing learning outcomes beyond those of visual imagery alone [49, 50].

The transfer test results revealed no significant difference between visual and audiovisual imagery. It is important to note that the transfer test assesses an individual's ability to adapt to new conditions, serving as an indicator of learning ability. The lack of difference observed in this study between audiovisual and visual imagery during the transfer test suggests that the level of mental and physical practice undertaken did not sufficiently improve subjects' skills to a level necessary for effective performance in different conditions.

The subjects, who engaged in both physical and mental practice from only one side of the court, likely did not develop the necessary level of skill transfer needed to perform effectively when faced with changes in court layout and angles. This suggests that additional or varied practice conditions may be required to improve skill transfer and adaptability in different scenarios.

The results of this study also demonstrated that during the retention test, the performance of the AudioVisual-Internal group significantly outperformed other groups, suggesting that internal imagery effectively activates the motor and kinesthetic senses. According to the input and output processing hypotheses, muscle stimulation occurs covertly during motor imagery [51]. Therefore, the benefits of motor imagery can be explained through these hypotheses. Input processing suggests that motor imagery activates muscular and peripheral structures, providing proprioceptive feedback to the central nervous system. On the other hand, output processing suggests that motor imagery induces changes in the central motor program, with observed electromyography (EMG) activity serving as evidence of imagery's impact and benefits [52]. Moreover, studies have indicated higher levels of EMG activity during motor imagery using internal visual imagery [53, 54]. Rodgers et al. [55] proposed that internal imagery is particularly effective in creating motor images compared to external imagery.

In this study, the velocity of the shoulder joint angle was converted into sound and presented as an auditory pattern. This auditory pattern provided subjects with information about motor kinematics and the nature of the movement itself. Therefore, during AudioVisual-Internal imagery, combining auditory-motor information (sonification of shoulder joint angular velocity) with internal imagery (strong potential for creating kinesthetic images or a sense of

motor) resulted in increased muscle involvement, which was evident in actual performance. These findings align with Ramzanzade et al.'s [36] study, which showed that muscle activity intensity is higher during AudioVisual-Internal and AudioVisual-Kinesthetic imagery compared to Visual imagery conditions (internal or kinesthetic), but no such difference was observed with AudioVisual-External imagery. Ramzanzade et al. [36] also demonstrated a relationship between the ability of internal and kinesthetic imagery. However, these results differ from those of Lutz [51], who suggested that muscle activity is a by-product of brain processing and does not contribute to the acquisition and retention of motor performance. Lutz's study [51] indicated that although motor imagery increases muscle activation, it does not impact acquisition and learning. In contrast to Lutz's argument, according to the input processing hypothesis, imagery leading to increased muscle activity provides sensory feedback to the brain, potentially helping acquisition and learning [52].

Finally, this study also showed that internal imagery, regardless of whether it is visual or audiovisual, has a greater impact on performance during retention and transfer tests compared to external imagery. This finding is particularly relevant for the badminton long serve, which is classified as a closed skill, suggesting that closed skills benefit from a stronger focus on internal imagery [56, 57]. This relationship likely stems from the ability of internal imagery to improve the motor sense crucial for representing closed skills [58, 59]. These findings align with previous research on closed skills such as dart throwing [56, 60], tennis serving [61], badminton serving [58] and football passing [12], all of which highlight the effectiveness of internal imagery. Studies have also shown that subjects using imagery from an external perspective record greater levels of EMG activity during MI [53, 54].

In this study focused on learning the badminton long serve, the impact of audiovisual imagery was investigated. Although no significant differences were observed during the acquisition stage between groups, participants in the audiovisual imagery condition demonstrated superior performance during the retention test compared to those in the visual imagery condition. Additionally, during the transfer test, the internal audiovisual group exhibited significantly higher performance than both the internal visual and external visual groups. These results indicate that using audiovisual imagery based on motor sonification helps skill development more effectively than visual imagery alone through practice. Furthermore, the AudioVisual-Internal imagery group outperformed the other groups, likely due to the auditory pattern providing detailed information about skill mechanics (skill pattern) in alignment with the motor sense cultivated by internal imagery. This synergy likely resulted in imagery with higher vividness and control, translating to improved performance. Overall, the use of audiovisual imagery incorporating sonification appears to improve the quality of imagery and subsequently impact performance and learning. However, further research is warranted to examine this topic in more depth and refine our understanding.

## 7. Limitation

One of the most important limitations of this research is that only the result has been recorded and the execution process, which includes kinematic analysis, has not been recorded. It is suggested that the execution process be investigated in future studies.

## Supporting information

**S1 Appendix.**
(SAV)

## Acknowledgments

The authors would like to show appreciation to the participants in this research.

## Author Contributions

**Conceptualization:** Fateme Parimi, Hesam Ramezanzade.

**Data curation:** Hesam Ramezanzade.

**Investigation:** Fateme Parimi.

**Methodology:** Fateme Parimi.

**Project administration:** Fateme Parimi, Behrouz Abdoli, Hesam Ramezanzade.

**Supervision:** Behrouz Abdoli, Hesam Ramezanzade, Mahin Aghdaei.

**Writing – original draft:** Fateme Parimi.

**Writing – review & editing:** Fateme Parimi, Behrouz Abdoli, Hesam Ramezanzade, Mahin Aghdaei.

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
