## [Decision Letter · Decision Letter 0]

20 Mar 2024

PONE-D-23-43363The Effect of Internal and External Audiovisual Imagery on Learning Badminton Long Serve SkillPLOS ONE

Dear Dr. Parimi,

Thank you for submitting your manuscript to PLOS ONE. After careful consideration, we feel that it has merit but does not fully meet PLOS ONE’s publication criteria as it currently stands. Therefore, we invite you to submit a revised version of the manuscript that addresses the points raised during the review process.

This manuscript has been assessed by 3 reviewers and their comments are available below. They feel the manuscript would benefit from thorough copyediting for proper English language usage and grammar, more methodological details and justification as well as improvements made to the discussion. Could you please carefully revise the manuscript to address all the comments made by the reviewers? 

We look forward to receiving your revised manuscript.

Kind regards,

Annesha Sil, PhD

Associate Editor, PLOS ONE

Reviewers' comments:

Reviewer's Responses to Questions

**Comments to the Author**

1. Is the manuscript technically sound, and do the data support the conclusions?

Reviewer #1: Yes

Reviewer #2: Yes

Reviewer #3: Partly

2. Has the statistical analysis been performed appropriately and rigorously? 

Reviewer #1: Yes

Reviewer #2: Yes

Reviewer #3: No

3. Have the authors made all data underlying the findings in their manuscript fully available?

Reviewer #1: Yes

Reviewer #2: Yes

Reviewer #3: No

4. Is the manuscript presented in an intelligible fashion and written in standard English?

Reviewer #1: Yes

Reviewer #2: No

Reviewer #3: No

5. Review Comments to the Author

Reviewer #1: Referring to PONE-D-23-43363

Title: The Effect of Internal and External Audiovisual Imagery on Learning Badminton Long Serve Skill

This study aims to investigate the effect of internal and external audiovisual imagery on learning badminton long serve skill. It is explores the influence of different motor imagery methods on motor learning and performance. It's a very interesting topic and valuable for educational implications in sport. The unique auditory model used in this research (based on which the characteristics kinematics of the movement has been converted into sound) and its use in motor imagery is the new approach of this research in motor imagery. Below are some suggestions that should be considered by the authors of the manuscript:

Abstract

In the abstract of the manuscript, the result of comparing the internal and external imagery groups should be reported.

Introduction

In the introduction, it is suggested to refer to the researches that have investigated the effect of sonification-based intervention on performance and motor learning.

Research Method

In line 202, refer to the source of sonification sandbox software

It is better to present the research protocol in the form of a table

Research findings

In the findings section, a number of effect sizes not reported. Be sure to report effect sizes for all pairwise comparison.

Discussion and conclusion

The discussion of this manuscript is well written and the explanation of the results is good based on previous researches and the proposed hypotheses. It seems that the effect of the intervention in this research is first on the quality of motor imagery and then on performance and motor learning. Therefore, it is better to mention this topic at the end of the discussion.

Reviewer #2: This manuscript aimed to evaluate the effects of internal and external audiovisual imagery on learning badminton long serve skill and the results showed that audio-visual internal imagery have a significant effect on learning badminton long serve skill. The manuscript suffers from major and minor problems as follows:

1. First of all, I need to point out that the language of the article is not appropriate and should be improved.

2. Please add the sampling method and statistical analysis methods to the abstract.

3. In the article, some terms are not used correctly, it is necessary to check all specialized terms again. For example, in some places you should use performance instead of execution.

4. The background section in the introduction is not well written and the results of some consecutive studies are reported. It is suggested to be rewritten.

5. In the last paragraph of the introduction, use retention instead of “retetion”. Check the spelling of words throughout the article.

6. Why did you use novice participants in the present study? Is it effective to use imagery at this stage and before learning the motor pattern?

7. In the method section, it is necessary to mention the sampling method and the method of determining the sample size.

8. Did you use the statistical method of mixed design ANOVA in the acquisition phase? It seems that the analysis of 4 groups * 5 stages of the test with repetition of the last factor has been used.

9. It is suggested to remove the tables and report the results in text form.

10. The quality of the figures in the article is not suitable. It is suggested to improve the quality of the presentation and to mention the error bar and significant differences in the figures.

11. In the first paragraph of the discussion, only mention the purpose and results. It is not necessary to mention all the details of the method of research.

12. The discussion section needs to be reorganized. It is necessary to compare the results of the study with previous studies and then explain the findings.

Reviewer #3: The work entitled "The Effects of Internal and External Audio-Visual Imagery on Learning Badminton Long Serve Skill" aimed to investigate the effects of motor imagery on motor learning. The study compared four conditions derived from the association of two independent variables: internal and external motor imagery, and visual and audio-visual imagery groups. The authors identified that there was no significant difference among groups during practice. However, the audio-visual-internal imagery group demonstrated better performance in the retention and transfer tests. Unfortunately, due to some inconsistencies in the rationale, internal coherence (rationale, methods, result discussion) analysis, and discussion, I suggest the non-acceptance of the work for publication. However, I believe that if the authors address the following topics, they could have a better outcome in further submission. I hope that my feedback can help them.

It has been noted that there are some mistakes in the manuscript, and it is recommended that the English language be assessed by a native speaker.

Title: Authors should focus on their two independent variables, namely internal versus external motor imagery and audiovisual versus visual motor imagery. The title seems to focus only on internal x external.

Introduction:

Line 52: Learning is not related to lifelong processes in the Motor Behavior area. Instead, Motor Development studies investigate motor behavior across the lifespan.

Line 54: Please use the traditional motor learning concept, "a series of processes associated with practice or experience that led to a relatively permanent change in the capacity to perform motor skills." I suggest reading "Motor Learning and Control" by Schmidt to verify why the words in the concept are important.

Line 57: What exactly does "accelerate learning" mean? Does it refer to acquiring motor skills quickly? Definitely, it is not assessed in motor learning. Maybe enhancing the process is a better alternative.

Line 61: The authors did not have neuroimage data to introduce or discuss neurophysiological mechanisms. Classic motor learning studies provide sufficient evidence and mechanisms at the behavioral level of analysis to support this study.

Line 65: Psycho Neuromuscular Theory is not the best option for the mechanisms that could influence motor learning by motor imagery. Please revisit classic books on motor learning (Schmidt, Magill, etc.).

Line 67: Avoid long paragraphs like this one as they complicate the reader's understanding.

Line 110: Again, keep the mechanisms at the behavioral level, given that the authors do not have data about neuroexcitability.

Line 121: It is a little confusing as the introduction does not guide the reader to investigate the inclusion of audio in visual imagery from internal and external perspectives. For me, the introduction needs to be rewritten to focus specifically on the gap in the literature regarding internal versus external and audio versus audiovisual imagery. The rationale is not strong enough and needs to be recreated.

Research Methods

Sample Size Calculation:

The manuscript lacks information on sample size calculation. Authors are encouraged to provide details regarding how the sample size was determined for the study.

Line 128: Clarify how each inclusion criterion was assessed. For instance, specify the methods used to confirm that participants were right-handed and free from neurological, hearing, visual (mention if a visual acuity test was employed), motor (explain the assessment method), attention (mention if MoCA - Montreal Cognitive Assessment - was utilized), and depression disorders (indicate if Beck Depression Scale was administered).

Line 129:- The English needs improvement. A suggested revision is: "All participants provided written consent to participate in the research."

Line 206:- Consider using the term "post-test" instead of "acquisition test." In general, in motor learning studies acquisition phase refers to phase across pre-test, blocks of practice and post-test.

Line 210): Include additional information in the inclusion criteria, such as: "Participants who scored above 70% on the mental rotation test were selected to participate in the research."

Lines 222-239: The description appears to be redundant and repetitive. Authors are advised to revise and streamline this section for clarity and conciseness.

Line 235: Provide a clear explanation of the terms "internal" and "external." Consider introducing this information in the introduction section for better contextual understanding.

Line 44 and Throughout Manuscript: Avoid using "attempt"; use "trial" instead. Ensure consistency in language usage across the entire manuscript.

Statistical Methods:

What exactly was the measure? Radial error? Mean error? Variable error? Please revisit Schmidt et al 2019, or Magill e Anderson 2021, to ensure the measures are ok.

The authors explain the analyses here, but they describe the analyses in the results section again. Please be sure that all explanations about the analyses (and how it align with your rational and aims) are explained here, just here.

Research Findings:

I strongly recommend considering the incorporation of error bars in your graphs. This addition is crucial for a comprehensive analysis of your results, providing valuable insights into the variability and precision of the data.

It is essential to ensure a standardized use of colors across groups in your line graphs. Consistency in color representation enhances visual clarity and facilitates accurate interpretation.

Authors might consider avoiding the inclusion of a table with ANOVA results. Providing a descriptive summary in the text and using line graphs to indicate statistical significance could be sufficient for clarity and conciseness.

I recommend reconsidering the reanalysis of data based on the internal and external, and audiovisual x visual factors. It is crucial to address potential confounding variables appropriately. Analyzing subgroups together might not be suitable when there are two independent variables influencing the outcome. Separate analyses for each variable could provide more accurate insights.

Performing a two-way ANOVA (post-test x retention; groups) may reveal essential aspects about the consolidation effect of imagery. This approach allows for a more nuanced understanding of how groups retain, lose, or enhance their improvements during the consolidation phase, potentially providing richer insights compared to a one-way ANOVA in retention or transfer alone.

Discussion Enhancement Response:

The first two paragraphs in the Discussion section should be removed. There is no need to reiterate methodological details in this section.

- It's important to keep the discussion focused on the behavioral level since no neurophysiological measures were employed.

- Emphasize that the audio-visual intervention did not show an effect during the acquisition phase; its impact was consolidation-dependent. See Kantak and Winstein (2012) to provide a framework for understanding the phases of motor memory affected by your intervention.

Line 373: You do not have cognitive effort measure to state any intervention effect on that during practice. Please avoid this extrapolation.

- Ensure accuracy in referencing Challenge Point Hypothesis to avoid confusion. Challenge Point is not related to your study object.

- Clarify the distinction between learning plateau and performance plateau in line 398. Use the appropriate term to convey the intended meaning.

- Acknowledge the speculative nature of the sentence in line 404, given that the data do not support the inference made.

- Your study is not a implicit motor learning study (line 416).

- Ensure consistency and accuracy in referencing Schmidt's name throughout the text (there is another author with similar name).

- Clearly differentiate between Gibson's Ecological Approach and mirror neuron theory (information processing). Ensure that inferences align with a coherent theory of motor behavior. There is no option to speculate the effect of your intervention using the two theories simultaneously.

- Break down the lengthy paragraph at 410 into more digestible sections for improved readability.

- Clearly state that the absence of a sample size calculation is a limitation, as opposed to considering the number of participants a limitation. Kinematic measure absence is not a limitation; it is a characteristic.

6. PLOS authors have the option to publish the peer review history of their article (what does this mean?). If published, this will include your full peer review and any attached files.

Reviewer #1: No

Reviewer #2: No

Reviewer #3: **Yes: **Giordano Marcio Gatinho Bonuzzi

---

## [Author Response · Author response to Decision Letter 0]

21 May 2024

Re: The Effect of Internal and External Audiovisual Imagery on Learning Badminton Long Serve Skill: The role of visual and audiovisual imagery

Dear Editor, 

Thank you for the opportunity to revise our manuscript. We have greatly appreciate the time and effort put forth by the reviewers, whose insightful comments have allowed us to significantly improve the quality of our work. Below, we have provided point-by-point responses to the reviewers' comments and have highlighted the corresponding changes made to the manuscript in blue font. We hope that the revised manuscript now meets the journal's standards for publication and look forward to your decision.

Sincerely, 

The Authors

Reviewer 1:

General comments

This study aims to investigate the effect of internal and external audiovisual imagery on learning badminton long serve skill. It is explores the influence of different motor imagery methods on motor learning and performance. It's a very interesting topic and valuable for educational implications in sport. The unique auditory model used in this research (based on which the characteristics kinematics of the movement has been converted into sound) and its use in motor imagery is the new approach of this research in motor imagery. Below are some suggestions that should be considered by the authors of the manuscript:

Response:

We are grateful for the reviewer's detailed feedback, which has proven invaluable in enhancing the quality of our manuscript. We sincerely appreciate the time and effort dedicated to reviewing our work.

Specific comments

Abstract

In the abstract of the manuscript, the result of comparing the internal and external imagery groups should be reported. 

Response: Thank you for the suggestion. The result of comparing the internal and external imagery groups has been included in the revised abstract (Line 44-46).

Introduction

In the introduction, it is suggested to refer to the researches that have investigated the effect of sonification-based intervention on performance and motor learning.

Response: Thank you for the comment. Lines 112 -116 now mention relevant studies.

Research Method

In line 202, refer to the source of sonification sandbox software. 

Response: Thank you for the thoughtful comment. The source of the sandbox software has been added. Lines 222, 556.

Walker BN, Cothran JT. Sonification Sandbox a graphical toolkit for auditory graphs. Proceedings of the International Conference on Auditory Display; 2003; Boston MA, USA. p. 6-9.

It is better to present the research protocol in the form of a table 

Response: Thank you for the comment. A research protocol table has been included in the revised manuscript. (Figure 2) 

Research findings

In the findings section, a number of effect sizes not reported. Be sure to report effect sizes for all pairwise comparison. 

Response: Thank you for the careful comment. The findings section has been completed. Lines 263-314.

Discussion and conclusion

The discussion of this manuscript is well written and the explanation of the results is good based on previous researches and the proposed hypotheses. It seems that the effect of the intervention in this research is first on the quality of motor imagery and then on performance and motor learning. Therefore, it is better to mention this topic at the end of the discussion.

Response: Thank you for the attention. This topic has now been mentioned. Lines 451 and 452.

Reviewer 2:

General comments

This manuscript aimed to evaluate the effects of internal and external audiovisual imagery on learning badminton long serve skill and the results showed that audio-visual internal imagery have a significant effect on learning badminton long serve skill. The manuscript suffers from major and minor problems as follows:

Response:

We sincerely appreciate the reviewer's detailed comments, which have significantly enhanced the paper's quality. We value the ability to identify various aspects of the article and provide valuable Feedback.

Specific comments

1. First of all, I need to point out that the language of the article is not appropriate and should be improved.

Response: Thank you for the careful reading. As suggested, the paper has been reviewed and revised by a native English speaker. Several edits and corrections have been made in the main text to improve readability.

2. Please add the sampling method and statistical analysis methods to the abstract. 

Response: Thank you for the comment. The sampling method and statistical analysis methods have been added to the abstract (Lines 32 and 40).

3. In the article, some terms are not used correctly, it is necessary to check all specialized terms again. For example, in some places you should use performance instead of execution.

Response: Thanks for the notice. The correction has been made.

4. The background section in the introduction is not well written and the results of some consecutive studies are reported. It is suggested to be rewritten.

Response: Thank you for the comment. The introduction has been completely revised, and research hypotheses have been added to the end of the introduction.

5. In the last paragraph of the introduction, use retention instead of “retetion”. Check the spelling of words throughout the article.

Response: Thank you for the careful comment. The correction has been made.

6. Why did you use novice participants in the present study? Is it effective to use imagery at this stage and before learning the motor pattern?

Response: Thank you for the comment.

 Imagery has two important functions: motivational and educational. The educational function includes techniques and tactics (strategies) practice. In the techniques practice function, the main purpose is to teach the motor pattern to novices. Many studies in motor behavior have investigated the effects of interventions on novices whit no prior experience who are in the cognitive stage of learning. Some examples of these studies are:

Driskell, J., Copper, C., & Moran, A. (1994). Does mental practice enhance performance. Journal of Applied Psychology, 79, 481-492.

Carien, R., Taktek, K., Hatchi, V., & Dominique, L. (2023). Effect of motor imagery training on service performance in novice tennis players: the role of imagery ability. International Journal of Sport and Exercise Psychology, 1–13. . 

Frank, C. (2016). Learning a motor action "From Within": Insight into perceptual-cognitive changes with mental and physical practice. Sport and Exercise Psychology Research, 91-121. 10.1016/B978-0-12-803634-1.00005-4

7. In the method section, it is necessary to mention the sampling method and the method of determining the sample size.

Response: Thank you. In this research available sampling was used. Lines 143, 145-147.

The G * power software was used to calculate the sample size for the ANOVA repeated measures test (α = .05; β= .95, group number = 4, number of measurement= 5, effect size = .223) (Pourmorad Kohan, Hatami & Baghaiyan, 2016), one way ANOVA and two way ANOVA (α = .05; β= .85, group number = 4, effect size = .58) (Smith & Holmes, 2004; Marshall & Gibson, 2017). 

Smith, D., & Holmes, P. (2004). The effect of imagery modality on golf putting performance. Journal of Sport and Exercise Psychology, 26(3), 385-395.

Marshall, E. A., & Gibson, A. M. (2017). The effect of an imagery training intervention on self-confidence, anxiety and performance in acrobatic gymnastics–a pilot study. Journal of Imagery Research in Sport and Physical Activity, 12(1), 20160009.

Pour Morad Kohan P, Hatami F, Baghaiyan M. The Effects of Sensory Modalities of Mental Imagery on Learning Lay-up Shot in Basketball. Motor Behavior. 2016;8 (26):173-88.

8. Did you use the statistical method of mixed design ANOVA in the acquisition phase? It seems that the analysis of 4 groups * 5 stages of the test with repetition of the last factor has been used.

Response: We conducted a factorial analysis (4 groups x 5 stages) to compare group performance during the acquisition stages. One-way ANOVA was used to compare group performance on the retention and transfer tests. Additionally, two-way ANOVA examined the interactive effects of imagery type (visual, audiovisual) and perspective (internal, external) on retention and transfer test performance.

9. It is suggested to remove the tables and report the results in text form.

Response: Thank you for your comment. We have removed the tables and instead reported the results in written form within the text. 

10. The quality of the figures in the article is not suitable. It is suggested to improve the quality of the presentation and to mention the error bar and significant differences in the figures.

Response: Thank you for your careful comment. The figures have been revised accordingly.

11. In the first paragraph of the discussion, only mention the purpose and results. It is not necessary to mention all the details of the method of research.

Response: Thank you for your comment. The extraneous details have been removed. (Lines 317-322)

12. The discussion section needs to be reorganized. It is necessary to compare the results of the study with previous studies and then explain the findings.

Response: Thank you for your careful comment. We have incorporated some useful changes in the discussion section.

Reviewer 3:

General comments

The work entitled "The Effects of Internal and External Audio-Visual Imagery on Learning Badminton Long Serve Skill" aimed to investigate the effects of motor imagery on motor learning. The study compared four conditions derived from the association of two independent variables: internal and external motor imagery, and visual and audio-visual imagery groups. The authors identified that there was no significant difference among groups during practice. However, the audio-visual-internal imagery group demonstrated better performance in the retention and transfer tests. Unfortunately, due to some inconsistencies in the rationale, internal coherence (rationale, methods, result discussion) analysis, and discussion, I suggest the non-acceptance of the work for publication. However, I believe that if the authors address the following topics, they could have a better outcome in further submission. I hope that my feedback can help them.

Response:

Thank you for taking the time and effort to review our manuscript. We have carefully considered each of your specific comments and addressed them as follows:

Specific comments

It has been noted that there are some mistakes in the manuscript, and it is recommended that the English language be assessed by a native speaker. 

Response: Thank you for your careful reading. As you suggested, the paper has undergone review and revision by a native English speaker. Several edits and corrections have been implemented throughout the main text to enhance readability.

Title

Authors should focus on their two independent variables, namely internal versus external motor imagery and audiovisual versus visual motor imagery. The title seems to focus only on internal x external.

Response: Per the reviewer's comment, the title of the manuscript has been modified. (Line 6)

Introduction

Line 52: Learning is not related to lifelong processes in the Motor Behavior area. Instead, Motor Development studies investigate motor behavior across the lifespan.

Response: Thank you for your comment. The issue you identified has been corrected. (Line 54).

Line 54: Please use the traditional motor learning concept, "a series of processes associated with practice or experience that led to a relatively permanent change in the capacity to perform motor skills." I suggest reading "Motor Learning and Control" by Schmidt to verify why the words in the concept are important.

Response: Thank you for your comment. The definition of motor learning in the manuscript has been revised accordingly (Lines 56-57).

Line 57: What exactly does "accelerate learning" mean? Does it refer to acquiring motor skills quickly? Definitely, it is not assessed in motor learning. Maybe enhancing the process is a better alternative.

Response: Thank you for your comment. The phrase "enhancing the process" has been replaced (Line 60).

Line 61: The authors did not have neuroimage data to introduce or discuss neurophysiological mechanisms. Classic motor learning studies provide sufficient evidence and mechanisms at the behavioral level of analysis to support this study.

Response: Thank you for your insightful feedback. Since the present study's underlying logic is based on Ramezanzade et al. (2023) research, which employed a physiological approach, the explanations provided in our research are also grounded in this physiological perspective. Ramezanzade et al. (2023) demonstrated that audiovisual imagery leads to a greater increase in muscle activity amplitude compared to visual imagery alone.

However, your comment is well-taken, and we have endeavored to incorporate a behavioral approach in our explanations within the introduction and discussion sections.

Line 65: Psycho Neuromuscular Theory is not the best option for the mechanisms that could influence motor learning by motor imagery. Please revisit classic books on motor learning (Schmidt, Magill, etc.).

Response: Thank you for your helpful comment. We have added discussions of other relevant theories to address this point. (Lines 69-75)

Line 67: Avoid long paragraphs like this one as they complicate the reader's understanding.

Response: Thank you for your comment. We have tried to make the sentence more concise. Line 76.

Line 110: Again, keep the mechanisms at the behavioral level, given that the authors do not have data about neuroexcitability.

Response: As mentioned in our previous response, the underlying logic of our research hypothesis is based on studies that have shown that audiovisual imagery can increase muscle activity amplitude more than visual imagery alone. Consequently, we have employed neurological explanations in the discussion and conclusion sections. However, your opinion is valid, and we have made efforts to incorporate a behavioral approach in our explanations within the introduction and problem statement sections.

Line 121: It is a little confusing as the introduction does not guide the reader to investigate the inclusion of audio in visual imagery from internal and external perspectives. For me, the introduction needs to be rewritten to focus specifically on the gap in the literature regarding internal versus external and audio versus audiovisual imagery. The rationale is not strong enough and needs to be recreated.

Response: Thank you for your careful comment. In line with your suggestion, we have added definitions of internal and external imagery to the introduction. Additionally, we have revised the introduction section thoroughly. Moreover, we have included the research hypotheses at the end of the introduction. (Lines 83-86)

Research Methods

Sample Size Calculation:

The manuscript lacks information on sample size calculation. Authors are encouraged to provide details regarding how the sample size was determined for the study.

The G * power software was used to calculate the sample size for the ANOVA repeated measures test (α = .05; β= .95, group number = 4, number of measurement= 5, effect size = .223) (Pourmorad Kohan, Hatami & Baghaiyan, 2016), one way ANOVA and two way ANOVA (α = .05; β= .85, group number = 4, effect size = .58) (Smith & Holmes, 2004; Marshall & Gibson, 2017). Lines 145-147.

Smith, D., & Holmes, P. (2004). The effect of imagery modality on golf putting performance. Journal of Sport and Exercise Psychology, 26(3), 385-395.

Marshall, E. A., & Gibson, A. M. (2017). The effect of an imagery training intervention on self-confidence, anxiety and performance in acrobatic gymnastics–a pilot study. Journal of Imagery Research in Sport and Physical Activity, 12(1), 20160009.

Pour Morad Kohan P, Hatami F, Baghaiyan M. The Effects of Sensory Modalities of Mental Imagery on Learning Lay-up Shot in Basketball. Motor Behavior. 2016; 8(26):173-88.

Line 128: Clarify how each inclusion criterion was assessed. For instance, specify the methods used to confirm that p

---

## [Decision Letter · Decision Letter 1]

10 Jul 2024

PONE-D-23-43363R1The Effect of Internal and External Imagery on Learning Badminton Long Serve Skill: The role of visual and audiovisual imageryPLOS ONE

Dear Dr. Parimi,

Thank you for submitting your manuscript to PLOS ONE. After careful consideration, we feel that it has merit but does not fully meet PLOS ONE’s publication criteria as it currently stands. Therefore, we invite you to submit a revised version of the manuscript that addresses the points raised during the review process.

We look forward to receiving your revised manuscript.

Kind regards,

M. Shamim Kaiser, PhD

Academic Editor

PLOS ONE

Additional Editor Comments:

NA

Reviewers' comments:

Reviewer's Responses to Questions

**Comments to the Author**

1. If the authors have adequately addressed your comments raised in a previous round of review and you feel that this manuscript is now acceptable for publication, you may indicate that here to bypass the “Comments to the Author” section, enter your conflict of interest statement in the “Confidential to Editor” section, and submit your "Accept" recommendation.

Reviewer #1: All comments have been addressed

Reviewer #2: All comments have been addressed

2. Is the manuscript technically sound, and do the data support the conclusions?

Reviewer #1: Yes

Reviewer #2: Yes

3. Has the statistical analysis been performed appropriately and rigorously? 

Reviewer #1: Yes

Reviewer #2: Yes

4. Have the authors made all data underlying the findings in their manuscript fully available?

Reviewer #1: Yes

Reviewer #2: Yes

5. Is the manuscript presented in an intelligible fashion and written in standard English?

Reviewer #1: Yes

Reviewer #2: Yes

6. Review Comments to the Author

Reviewer #1: All of my concerns are addressed by the authors. And therefore, I have no other comments in this stage.

Reviewer #2: (No Response)

7. PLOS authors have the option to publish the peer review history of their article (what does this mean?). If published, this will include your full peer review and any attached files.

Reviewer #1: No

Reviewer #2: **Yes: **Jalil Moradi

---

## [Author Response · Author response to Decision Letter 1]

18 Jul 2024

PONE-D-23-43363R1

The Effect of Internal and External Imagery on Learning Badminton Long Serve Skill: The role of visual and audiovisual imagery

Dear Academic Editor 

Dr Shamim Kaiser

I hope you are doing well.

We carefully read the letter of the dear academic editor. Both reviewers have approved the manuscript and there were no comments for revision. In the "view attachment" section, there is a file related to the previous stage of the revising process, and all the things mentioned in the file (related to reviewer 1) have already been done.

If what the academic editor meant by "Major Revision" is to upload the figures through the Preflight Analysis and Conversion Engine (PACE) digital diagnostic tool, we did this and used this tool to upload the figures when submitting the manuscript.

If there is anything else we need to do besides using PACE tool, please let us know.

At the time of submission, we sent the file of referee corrections related to the previous step and also the manuscript.

Best,

Fateme Parimi

---

## [Editor Report · Decision Letter 2]

13 Aug 2024

The Effect of Internal and External Imagery on Learning Badminton Long Serve Skill: The role of visual and audiovisual imagery

PONE-D-23-43363R2

Dear Dr. Parimi,

We’re pleased to inform you that your manuscript has been judged scientifically suitable for publication and will be formally accepted for publication once it meets all outstanding technical requirements.

Kind regards,

M. Shamim Kaiser, PhD

Academic Editor

PLOS ONE

Additional Editor Comments (optional):

Please read the paper very carefully to eliminate typos.

Improve the quality of the figures
---

## [Editor Report · Acceptance letter]

10 Sep 2024

PONE-D-23-43363R2 

PLOS ONE

Dear Dr. Parimi, 

I'm pleased to inform you that your manuscript has been deemed suitable for publication in PLOS ONE. Congratulations! Your manuscript is now being handed over to our production team.

Kind regards, 

on behalf of

Dr. M. Shamim Kaiser 

Academic Editor

PLOS ONE